# A Compact Stacked RF Energy Harvester with Multi-Condition Adaptive Energy Management Circuits

**DOI:** 10.3390/mi14101967

**Published:** 2023-10-22

**Authors:** Xiaoqiang Liu, Mingxue Li, Xinkai Chen, Yiheng Zhao, Liyi Xiao, Yufeng Zhang

**Affiliations:** School of Aeronautics, Harbin Institute of Technology, Harbin 150001, China; 20b921030@stu.hit.edu.cn (X.L.); li_mingxue@hit.edu.cn (M.L.); 15069301390@126.com (X.C.); 22s136086@stu.hit.edu.cn (Y.Z.); xiaoly@hit.edu.cn (L.X.)

**Keywords:** RF energy harvesting, rectenna, compact design, micro energy, energy management

## Abstract

This paper presents a compact stacked RF energy harvester operating in the WiFi band with multi-condition adaptive energy management circuits (MCA-EMCs). The harvester is divided into antennas, impedance matching networks, rectifiers, and MCA-EMCs. The antenna is based on a polytetrafluoroethylene (PTFE) substrate using the microstrip antenna structure and a ring slot in the ground plane to reduce the antenna area by 13.7%. The rectifier, impedance matching network, and MCA-EMC are made on a single FR4 substrate. The rectifier has a maximum conversion efficiency of 33.8% at 5 dBm input. The MCA-EMC has two operating modes to adapt to multiple operating conditions, in which Mode 1 outputs 1.5 V and has a higher energy conversion efficiency of up to 93.56%, and Mode 2 supports a minimum starting input voltage of 0.33 V and multiple output voltages of 2.85–2.45 V and 1.5 V. The proposed RF energy harvester is integrated by multiple-layer stacking with a total size of 53 mm × 43.5 mm × 5.9 mm. The test results show that the proposed RF energy harvester can drive a wall clock (30 cm in diameter) at 10 cm distance and a hygrometer at 122 cm distance with a home router as the transmitting source.

## 1. Introduction

With the development of communication and wireless sensing technology, electromagnetic waves are becoming more widely distributed in the environment. Electromagnetic waves are a kind of electromagnetic information as well as electromagnetic energy that has the advantages of being clean, wireless, wide distribution, etc. Harvesting ambient electromagnetic wave (radio frequency, RF) energy has become a contemporary research hotspot [1,2]. Compared to traditional ambient energy sources such as photovoltaic (PV), vibration, and wind, RF energy is widely distributed and insensitive to environmental factors (such as light intensity, vibration rate, wind speed, etc.). In addition, RF energy harvesters usually have a small volume and possess the advantages of integration [3,4].

WiFi is the most widely used wireless network-transmission technology today. By using wireless routers to send and receive electromagnetic waves, mobile phones and computers can achieve high-speed wireless data communication. WiFi band is divided into 2.4 G and 5 G, of which the 2.4 G band is 2.412–2.472 GHz in China. WiFi signal sources are generally omnidirectional to meet the signal demand in all directions. However, network signals are not required in all directions, resulting in the waste of RF energy. Placing radio frequency (RF) energy harvesters in the WiFi signal distribution areas that have no signal demand to harvest WiFi RF energy and power small electronic devices or wireless sensor nodes (WSNs) is of great value for energy savings and environmental protection.

An RF energy harvester generally consists of an energy-receiving antenna, a rectifier, and energy management circuits [5]. The antenna and rectifier, usually referred to as the rectenna, are the core devices for RF energy harvesting. The energy management circuits realize energy storage, release, and voltage conversion [6]. Generally, the rectifier contains an impedance matching network to achieve impedance matching with the antenna and maximize the efficiencies of energy transmission. According to the Friis equation, increasing the gain of the receiving antenna can significantly improve the efficiency of RF energy transmission [7]. For a rectenna, a smaller size means greater application compatibility and flexibility [8,9]. Increasing resonance points is a common method to achieve multi-frequency energy harvesting without increasing the area. A dual-band compact rectenna with an overall dimension of 80 mm × 45 mm and a maximum gain of 4 dBi was proposed in [10]. An additional resonance point was achieved with a minimal increase in overall size by using two inductors and one capacitor. To achieve dual-frequency rectification, the impedance matching network occupied a relatively large area. Additionally, to enhance the gain of the monopole antenna, a substantial ground area was required. These two factors constrained the integration level of the rectenna. In some previous studies, matching-free rectennas were achieved by incorporating the rectifier impedance during the antenna design. A compact rectenna without impedance matching network operating at 2.45 GHz with an overall size of 70 mm × 70 mm × 20 mm was proposed in [11]. To achieve the matching-free effect, the designed antenna structure does not have a significant advantage in terms of gain. To enhance the gain, this article employed a reflector plate. While the gain was improved, the overall volume of the rectenna increased. Some studies have achieved compact rectennas through structural optimization. A 2.45 GHz compact rectenna with an overall dimension of only 24.9 mm × 8.6 mm and a 0.8 dBi peak gain was reported in [12]. This research significantly reduced the volume of the rectenna by designing the antenna structure and integrating the matching circuits into the PCB edge. However, this also resulted in a significant decrease in antenna gain and rectification efficiency. In the previous studies, the proposed rectennas still have plenty of optimization space in terms of gain and total size, and few studies have achieved verification of ambient RF energy harvesting.

Generally, the RF energy management circuits are simple in structure and have a boost function because ambient RF energy is extremely weak and the output voltage of the rectifier is not enough to drive the load directly [13]. BOOST circuits were constructed using discrete devices in [14,15]. In [16,17,18], an ultra-low power management chip was first used to boost the voltage, and then a DC-DC chip was used to convert and output the load voltage. In more studies, a single energy management chip was used to realize energy “store-and-release” and voltage boosting [7,19,20,21]. RF energy management efficiency is affected by operating conditions including input power, input voltage, output voltage, and the load. The adaptability of RF energy management circuits for multiple operating conditions and multiple voltage outputs was less studied in previous research, which may lead to the inability of load driving under multiple operating conditions and low energy conversion efficiencies.

In this study, a compact stacked RF energy harvester with multi-condition adaptive energy management circuits is proposed. The antenna is designed with a microstrip structure to enhance the gain and a circular slot in the ground plane to reduce the area. The RF energy harvester is integrated by multilayer stacking, which greatly reduces the area and size of the system. The proposed energy management circuit supports two operating modes (high efficiency mode and wide input range mode) and two voltage outputs (1.5 V, 2.85–2.45 V). The efficiency of the RF energy management circuit can be maximized by switching between two modes. The contributions of this paper are concluded as follows:(1)A novel compact rectenna with a stacked structure.(2)A novel multi-condition adaptive energy management circuit for RF energy supporting two operating modes and two voltage outputs.(3)We conducted ambient RF energy harvesting validation for the proposed RF energy harvester. The validation results indicate that the proposed RF energy harvester can drive a wall clock (30 cm in diameter) at 10 cm distance and a hygrometer at 122 cm distance with a home router as the transmitting source.

The remainder of this paper is organized as follows: Section 2 presents a comprehensive description of the complete design of the RF energy harvester. Section 3 provides the details of the ambient RF energy harvesting experiments. Section 4 discusses the results obtained as well as crucial design considerations. Section 5 concludes the paper.

## 2. Design of the RF Energy Harvester

### 2.1. Design of the Antenna

The rectenna proposed in this paper is based on a double-layer stacking structure. The upper board is an energy-receiving antenna based on a polytetrafluoroethylene (PTFE) substrate of 1.52 mm thickness. The lower board is a rectifier based on an FR4 substrate of 1.6 mm thickness. The antenna and rectifier were designed by High Frequency Structure Simulator (HFSS) and Advanced Design System (ADS), respectively.

The dimensions and photograph of the proposed antenna are shown in Figure 1. The dimensional parameters of the proposed antenna and rectifier are listed in Table 1. The antenna is composed of a radiating patch, a feedline, and a ground plane, as shown in Figure 1b,d. In Figure 1a, the resonant frequency of the antenna is determined by the dimensions (W0 and L0) of the radiating patch. The impedance matching of the 50 Ω feedline to the antenna patch is achieved by changing W2 and L2. Etching gaps in the ground plane to alter the antenna’s current transmission path is a common method for changing the antenna’s resonant frequency and dimensions. In Figure 1c,d, a circular slot is etched in the antenna ground plane to decrease the antenna resonant frequency. To achieve the original resonant frequency, it is necessary to reduce the size of the radiating patch, thus decreasing the total area. During the antenna design process, the circular slot alters the shape of the ground plane, which in turn changes the input impedance of the radiating patch. The change in impedance is influenced by the circular slot parameters R1 and R2. The parameters W2 and L2 used for impedance matching need to be adjusted accordingly. Therefore, by optimizing and adjusting the circular slot parameters, matching parameters, and the dimensions of the radiating patch in HFSS, antenna miniaturization and performance improvement are achieved. The antenna size is 56 mm × 47.7 mm without a circular slot and 53 mm × 43.5 mm with a circular slot, which means the slot decreases the total area by 13.7%.

Figure 2 shows the simulated and measured S11 results of the proposed antenna. The proposed antenna has a −10 dB bandwidth of 63 MHz (2.422–2.485 GHz) and a S11 of −19.77 dB at 2.452 GHz. Figure 3 shows 2D and 3D simulations of the radiation pattern. For the energy-harvesting antenna, the higher the gain, the stronger the directionality, and the higher the radiation efficiency. The proposed antenna has a maximum gain of 4.578 dBi and an omnidirectional radiation efficiency of 96.37%.

### 2.2. Design of the Rectifier

The rectifier and the impedance matching network are fabricated on an L-shaped FR4 substrate, as shown in Figure 4b,c. Figure 4a depicts a diagram of the rectifier and impedance matching network. The shape of the impedance matching network has been adjusted to accommodate the shape of the circular slot and avoid affecting the antenna’s performance. Due to the nonlinearity of the diode, the input impedance of the rectifier varies under different input power levels and different load resistances, which increases the design complexity of the impedance matching network. Figure 5 shows the input impedance of the rectifier under different input power levels (−30 dBm to 30 dBm) and load resistances (1000 Ω, 1500 Ω, 2000 Ω). It can be observed that the input impedance of the rectifier varies significantly at low input power levels, mainly because the diodes are not fully conducting at this point and their nonlinear characteristics become prominent. Common impedance matching methods include lumped element matching and microstrip line matching. It is difficult to achieve optimal matching using lumped components due to their discrete values. Microstrip-line matching allows for continuous dimension adjustment, resulting in continuous impedance variation and a better matching effect. Single-stub and double-stub are common microstrip-line matching structures. In this study, double-stub matching was chosen. In Figure 4b, L3 and L4 are 50 Ω feed lines, and L5 and L7 are the two impedance matching branches from 50 Ω to the rectifier. Impedance matching adjustments are achieved by optimizing the lengths of two stubs in ADS. The diode and isolation capacitor are HSMS2852 from Broadcom (Irvine, CA, USA) and 5.1 pF from Murata (Nagaokakyo, Japan), respectively. The simulation model for the diode is sourced from the high-frequency diode library in ADS. During the simulation process, the primary approach is to explore the highest conversion efficiency and globally higher efficiency by input power and load pull.

The measurement setup and a comparison of the simulated and measured RF-to-DC conversion efficiency of the proposed rectifier are shown in Figure 6 and Figure 7. In Figure 6, the RF signal source (DSG836 from RIGOL) outputs RF energy (−10 dBm–5 dBm) to the rectifier, and then the rectifier outputs to a resistance box (set to 1000 Ω). The voltage across the resistance box is measured by a multimeter (15 B+ from FLUKE). In Figure 7, the RF-to-DC conversion efficiencies were measured at different load resistances with the same input RF power of 5 dBm. It can be observed in Figure 6 and Figure 7 that the simulation and measurement results are generally consistent, and a maximum measured efficiency of 33.8% is achieved at 5 dBm input power and 1000 Ω load.

### 2.3. Design of the Multi-Condition Adaptive Energy Management Circuits

The structure of the proposed RF energy harvester equipped with multi-condition adaptive energy management circuits is shown in Figure 8. RF energy harvested by the rectenna is converted by the power management integrated chip (PMIC), BUCK, and Low Dropout Regulator (LDO) to Load1, Load2, and Load3, respectively. A double-pole-double-throw (dpdt) switch, together with Switch 1, Switch 2, and a hysteresis comparator is used to switch two operating modes. When the DPDT switch is toggled to positions 1 and 2, the system enters Mode 1 and Mode 2, respectively. In Mode 1, Load3 is supplied by the LDO, while the PMIC and BUCK serve as reference voltage sources. In Mode 2, Load1 is powered by the PMIC and LOAD2 is supplied by the BUCK, while the LDO is inactive. These two modes are designed to adapt to different input power and input voltage conditions, which can provide varying voltage ranges to accommodate various operating scenarios. The PMIC is BQ25504RGTT from Texas Instruments (Dallas, TX, USA), which is an ultra-low power management chip that enables voltage boosting at a minimum input voltage of 0.33 V [22]. BQ25504RGTT can output a threshold management signal. In this paper, the high and low voltage thresholds of BQ25504RGTT are set to 2.85 V and 2.45 V.

When the dpdt switch is adjusted to position 1 (Mode 1), the storage capacitor and the output of Switch 1 are connected to the recenna and Switch 2, respectively. As the voltage of the storage capacitor rises, Switch 2 is turned on, and the hysteresis comparator and the LDO are powered up. The output voltage of Switch 1 is converted by the BUCK to obtain a 1.5 V output as the supply and reference voltage (after divider) for the hysteresis comparator. The LDO step-down converts the output voltage of the rectenna and outputs 1.5 V to Load3. The threshold range of the hysteresis comparator is set to 1.5–1.8 V, i.e., when the storage capacitor voltage reaches 1.8 V, the LDO powers up and the capacitor discharges, and when the storage capacitor voltage is lower than 1.5 V, the LDO powers down and the capacitor stops discharging. The hysteresis comparator must have a stable reference voltage; otherwise, the threshold will be inaccurate because of the changing input voltage. Therefore, PMIC and BUCK are used as reference voltage generation sources in this paper. The power consumption of the reference voltage source includes only the static power consumption of the PMIC, BUCK, and some configuration resistors because Load1 and Load2 are not connected to the harvester in Mode 1.

When the dpdt switch is adjusted to position 2 (Mode 2), the storage capacitor is connected to the PMIC output, and Switch 1 no longer controls Switch 2. Switch 2 turns off, and the LDO and the hysteresis comparator are powered down. The storage capacitor acts as an energy storage element of the PMIC, and the PMIC monitors the voltage on it and controls Switch 1. In mode 2, the RF energy harvester can output 2.85–2.45 V and 1.5 V by connecting Load1 and Load2, respectively.

The power consumption and energy conversion efficiency of the MCA-EMC are affected by the load state, operating mode, input power, and input voltage, which are estimated in the next contents of this paper. The turn-off currents of the two switches (2nA), are neglected in the estimation. Equations (1)–(3) express the conversion efficiencies of the MCA-EMC when Load1, Load2, and Load3 are output individually, where $P_{in}$ denotes the input power of the MCA-EMC and $P_{qLDO}$ represents the power-off static power consumption of the LDO. R1, R2, R3, and R4 are the configuring resistors of the hysteresis comparator. Figure 9 illustrates the peripheral circuitry of the hysteresis comparator. $P_{R1+R2}$ denotes the power consumption of R1 and R2, whereas $P_{R3+R4}$ denotes the power consumption of R3 and R4. $P_{HComp}$ indicates the static power consumption of the hysteresis comparator. $\eta_{PMIC}$, $\eta_{BUCK}$, and $\eta_{LDO}$ represent the energy conversion efficiency of the PMIC, BUCK, and LDO, respectively. $P_{qBUCK}$ stands for the static power consumption of the BUCK.
(1)$$\eta_{LOAD1}=\frac{(P_{in}-P_{qLDO}-P_{R1+R2})\cdot \eta_{PMIC}-P_{qBUCK}}{P_{in}}$$
(2)$$\eta_{LOAD2}=\frac{(P_{in}-P_{qLDO}-P_{R1+R2})\cdot \eta_{PMIC}\cdot \eta_{BUCK}}{P_{in}}$$
(3)$$\eta_{LOAD3}=\frac{\left(P_{in}-P_{R1+R2}-P_{HComp}-\frac{P_{R3+R4}}{\eta_{BUCK}\cdot \eta_{PMIC}}\right)\cdot \eta_{LDO}}{P_{in}}$$

This paper conducted a simulation-based evaluation of energy conversion efficiency under two modes and three load outputs. Table 2 lists the leakage current, leakage power, and conversion efficiency (for power chips only) of each module. In the estimation, the range of $P_{in}$ is set from 10 µW to 1 mW. It should be noted that the power supply chip’s conversion efficiency varies with operating conditions. The efficiency data presented in this paper is the average values from the datasheet corresponding to the input power and load current ranges. The output voltage $V_{LOAD1}$ of LOAD1 is set at 2.85 V, while the output voltages $V_{LOAD2}$ and $V_{LOAD3}$ of LOAD2 and LOAD3 are both set at 1.5 V. The conversion efficiencies under three different loads were compared for input voltages $V_{in}$ of 1 V, 1.6 V, and 1.8 V, as shown in Figure 10. For LOAD1 and LOAD2, $V_{in}$ was chosen to be 1 V based on Equations (1) and (2), where lower $V_{in}$ results in lower $P_{qLDO}$ and $P_{R1+R2}$, leading to higher $\eta_{LOAD1}$ and $\eta_{LOAD2}$. Additionally, selecting Vin below 1.5 V better conforms to the operating conditions of LOAD1 and LOAD2. For LOAD3, the conversion efficiencies of the MCA-EMC at $V_{in}$ of 1.6 V and 1.8 V are compared. As shown in Figure 10, when the input voltage $V_{in}$ aligns with LOAD3’s operating conditions, LOAD3 exhibits a distinct efficiency advantage. For $P_{in}$ ranging from 60 μW to 1 mW, the efficiency exceeds 80% at $V_{in}$ of 1.8 V and surpasses 90% at $V_{in}$ of 1.6 V with the highest efficiency reaching 93.56%. If LOAD3’s input conditions are not met, it is necessary to switch to either LOAD1 or LOAD2 outputs. LOAD1 delivers a voltage output ranging from 2.85 V to 2.45 V, while LOAD2 maintains a consistent 1.5 V output. Due to a two-stage conversion process, LOAD2 has the lowest efficiency. LOAD1 and LOAD2 have lower demands on input voltage, making them suitable for a broader range of application scenarios.

Figure 11 illustrates the operational voltage changes in two modes. In Figure 11a, the threshold voltage of the hysteresis comparator is validated by adjusting $V_{in}$ to rise above 1.8 V and then drop below 1.5 V. During this process, VLOAD1 maintains a consistent maximum output voltage of 3.15 V, while VLOAD2 serves as a reference voltage and maintains a continuous output of 1.5 V. Figure 11b depicts the operational voltage state of Mode 2 with a $V_{in}$ of 1 V. The voltage across the capacitor rises to 2.85 V, causing the PMIC’s threshold monitoring pin to output a high level with an activation time of approximately 66 ms. In Mode 2, since the input voltage remains below 1.8 V, VLOAD3 remains at 0 V. For VLOAD1 and VLOAD2 in Mode 1 and VLOAD3 in Mode 2, only minimal static power consumption is generated because the subsequent stage is not connected to a load, resulting in minimal impact on energy conversion efficiency.

In this paper, the storage capacitor is shared in two modes by using a dpdt switch to reduce the system size. By coordinating components in two modes, energy charge and discharge control are achieved, further reducing energy management consumption and extending the input and output voltage range of the MCA-EMC. In Mode 1, the energy storage process operates with only a LDO chip, maximizing energy management efficiency.

## 3. Validation

The antenna, rectifier, and MCA-EMC were stacked and integrated as shown in Figure 12b to obtain the RF energy harvester in Figure 12d. The ground planes of the two parts are soldered together by tin. The antenna feed end and the rectifier feed end were connected by a SMA feed pin. Copper at the ground plane of both the antenna and the rectifier are spaced to ensure that there is no short between the feed and the ground during stacking. The total size of the proposed RF energy harvester is 53 mm × 43.5 mm × 5.9 mm, which is the same as the antenna in length and width.

Figure 13 shows the RF energy harvesting test. The RF signal source is DSG836 from RIGOL (Suzhou, China), the spectrum analyzer is N9030 A from Agilent Technologies, the transmitting antenna is a planar antenna with a gain of 14 dBi, and the receiving antenna is the antenna proposed in this study. The RF signal source emits RF energy through the transmitting antenna. The RF energy is received by the receiving antenna and measured by the spectrum analyzer. By moving the receiving antenna and measuring radiation power density at different antenna positions (measuring instrument: TES-92 from TES), the relationship between the radiation power density, the received energy and the distance between the transmitting antenna and the receiving antenna is established, as shown in Figure 14.

Figure 14 also illustrates the relationship between the radiation power density and the distance when a router is used as the transmission source. Because the signal emitted by the router is modulated and not continuous, it is challenging to quantify the received RF energy through spectrum analysis. However, in a stable network environment, radiation power density testing is measured in integral form. Therefore, ambient RF energy harvesting data can be quantified by comparing radiation power density. Figure 15 illustrates the ambient RF energy harvesting performance of the proposed RF energy harvester at various distances with a home router as the RF energy source. As depicted in Figure 15a, the minimum distance satisfying the operating conditions of Mode 1 is 30 cm. At this distance, the voltage output to LOAD3 is 1.849 V. From Figure 14, it can be observed that the radiation power density at the receiving antenna location is 80 µW/cm², corresponding to a received power of approximately −5 dBm. As shown in Figure 15b, the maximum distance for Mode 2 to output to LOAD1 and LOAD 2 is 122 cm. It can be seen from Figure 14 that the power received at the receiving antenna location is approximately −10 dBm. At this distance, LOAD1 can output a voltage above 2.85 V, and the storage capacitor has sufficient capacity to support a single data acquisition and display cycle for a hygrometer, as illustrated in Figure 15c. A wall clock of 30 cm in diameter, directly driven by a house router under Mode 1, is shown in Figure 15d. Upon connecting to Load1 in Mode 1, the output voltage decreases. A direct drive of the wall clock cannot be achieved at the 30 cm distance shown in Figure 15a. The test results show that the effective direct driving distance is approximately 10 cm. From Figure 14, it can be observed that the power received by the receiving antenna is approximately 3 dBm.

## 4. Discussion

This paper has realized a compact stacked RF energy harvester. In order to reduce costs and manufacturing complexity, FR4 material is chosen for producing the rectifier and energy management circuit. Opting for microwave materials would enhance the efficiency of the rectifier to a certain extent; however, it would correspondingly increase costs and manufacturing complexity.

Mode 1 requires an input open-circuit voltage of 1.8 V or higher, making it more suitable for scenarios with higher RF energy for direct driving. Mode 2 is adaptable to input voltages above 0.33 V, making it more suitable for the “store-and-release” scenario with lower RF energy. For Mode 1, due to threshold management designed in the circuit, it can also be used for “store-and-release” as long as its input conditions are met, yielding higher efficiency compared to Mode 2. The combined use of these two modes can maximize RF energy conversion efficiency. The 1.5 V output from both Mode 1 and Mode 2 can be applied to power small electronic devices, whereas the 2.85–2.45 V output from Mode 2 can be used to drive wireless sensor nodes. This paper did not achieve adaptive switching between the two modes, which will be a future research direction. The additional circuits introduced for adaptive switching may lead to additional power consumption issues, thereby increasing design complexity.

Compared to the previous studies, the RF energy harvester proposed in this paper has added circuits for energy management and mode switching, enhancing energy management efficiency and extending the output voltage range. In the previous studies, energy conversion efficiency was mostly focused on rectifier conversion efficiency (RF-DC efficiency) [7,16,17,19,21]. In comparison, this paper holds a substantial advantage. In the previous studies, RF energy harvesters generally consist of multiple interconnected modules, resulting in a larger overall size that was challenging to measure [10,23,24,25,26]. In contrast, this paper achieves stacking integration, presenting a compact square-shaped form that is well-suited for integrated applications. Table 3 lists the comparison of the proposed RF energy harvester and related designs.

## 5. Conclusions

A compact stacked RF energy harvester in the WiFi band is proposed. A circular slot is etched in the ground plane to reduce the total area of the antenna by 13.7%. The proposed rectenna has a maximum gain of 4.578 dBi, a S11 of −19.77 dB at 2.452 GHz, and a −10 dB bandwidth of 63 MHz. The proposed rectifier has a measured maximum RF-to-DC conversion efficiency of 33.8% at 5 dBm input. A multi-operating-condition adaptable energy management circuit (MCA-EMC) is proposed, supporting voltage inputs above 0.33 V and enabling outputs of 1.5 V and 2.85–2.45 V. Under 60 μW–1 mW power and a 1.6 V voltage input, the conversion efficiency of the MCA-EMC is above 90%, with a peak efficiency of 93.56%. The antenna, the rectifier, and the MCA-EMC are integrated by multi-layer stacking, with a total size of 53 mm × 43.5 mm × 5.9 mm. Ambient RF energy harvesting tests show that the proposed RF energy harvester achieves a maximum operating distance of 30 cm under Mode 1 and 122 cm under Mode 2 from a home router as the RF energy source. The compact size, high integration, multi-voltage outputs, and excellent conversion efficiency of the proposed RF energy harvester make it highly valuable for applications in ambient RF energy harvesting and integration into other devices.

## Figures and Tables

**Figure 1 micromachines-14-01967-f001:**
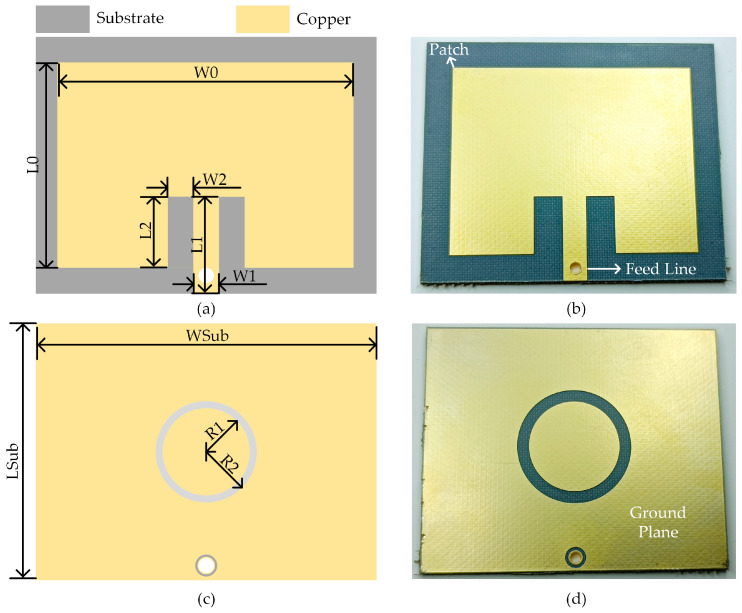
Antenna, (**a**) structure and dimension of the top layer; (**b**) photograph of the top layer; (**c**) structure and dimension of the bottom layer; (**d**) photograph of the bottom layer.

**Figure 2 micromachines-14-01967-f002:**
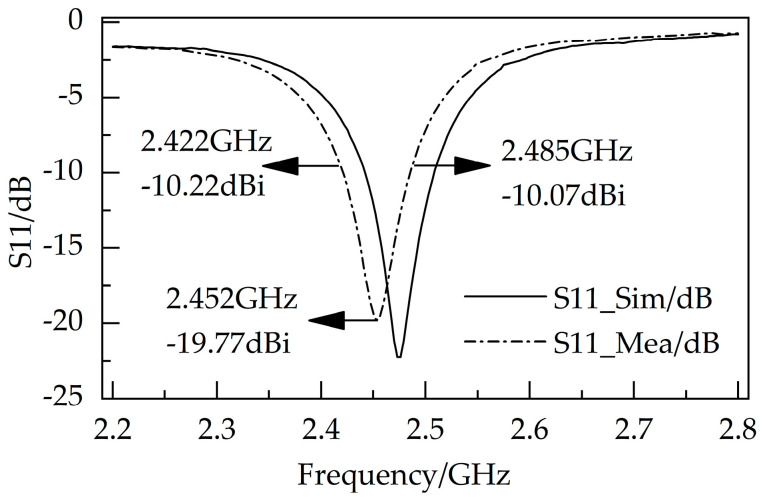
Simulated and measured S11 results of the proposed antenna.

**Figure 3 micromachines-14-01967-f003:**
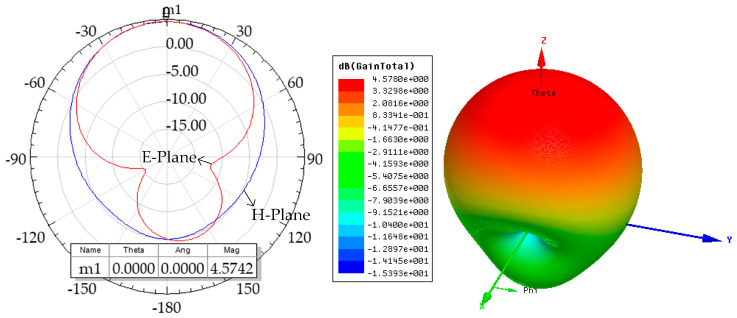
2D and 3D simulations of radiation patterns.

**Figure 4 micromachines-14-01967-f004:**
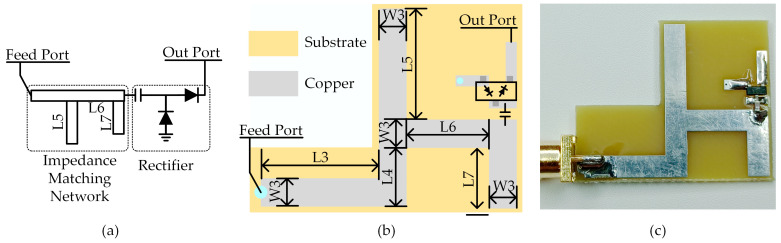
The impedance matching network and the rectifier, (**a**) diagram; (**b**) dimensions; (**c**) photograph.

**Figure 5 micromachines-14-01967-f005:**
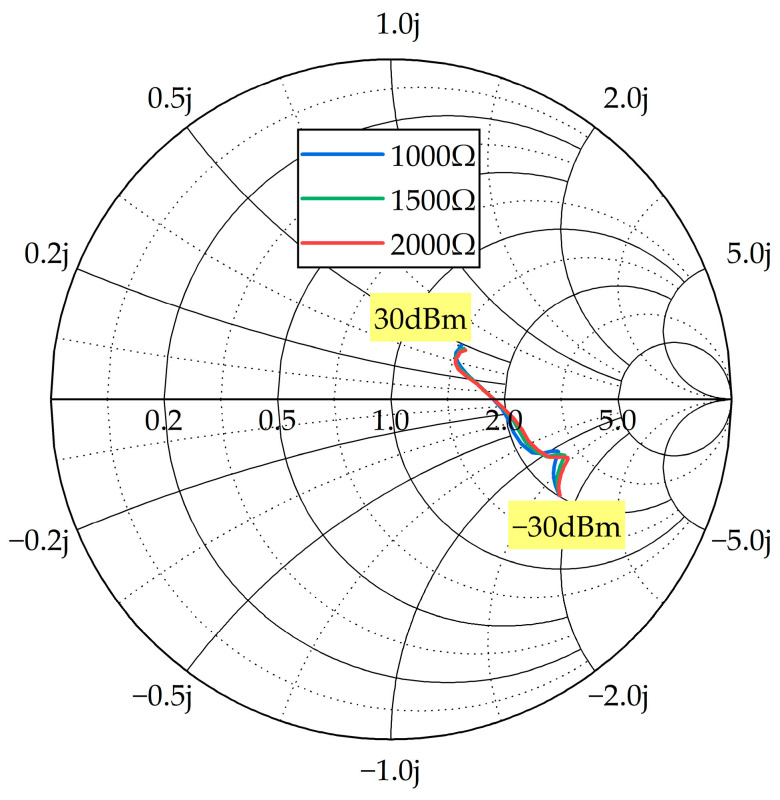
Input impedance of the rectifier at different input power levels and loads.

**Figure 6 micromachines-14-01967-f006:**
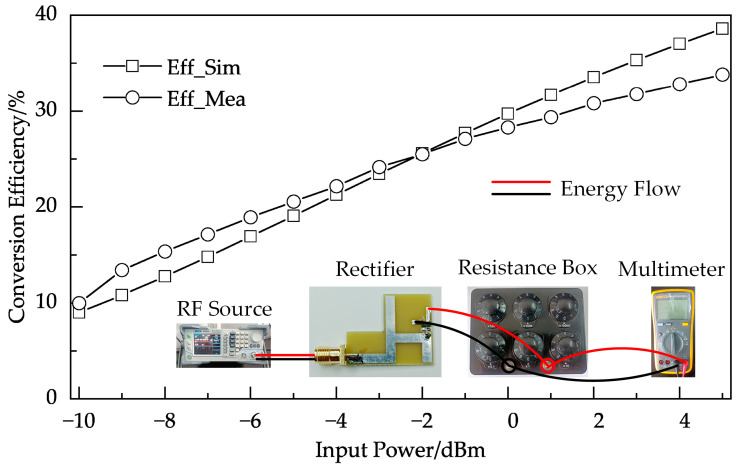
Measured and simulated RF-to-DC conversion efficiency of the proposed rectifier versus input power.

**Figure 7 micromachines-14-01967-f007:**
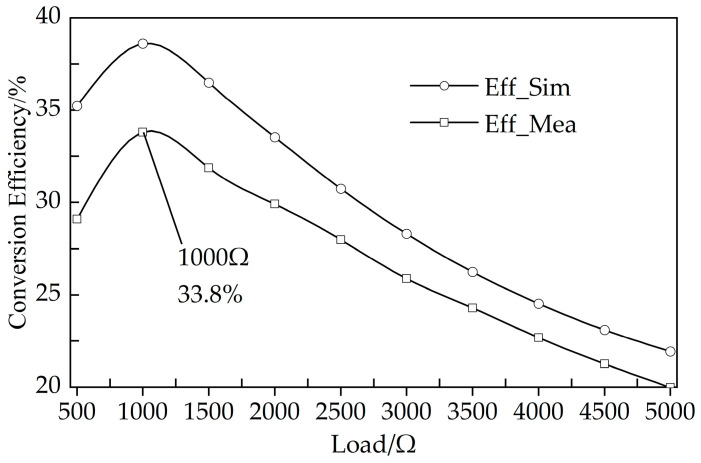
Measured and simulated RF-to-DC conversion efficiency of the proposed rectifier versus load.

**Figure 8 micromachines-14-01967-f008:**
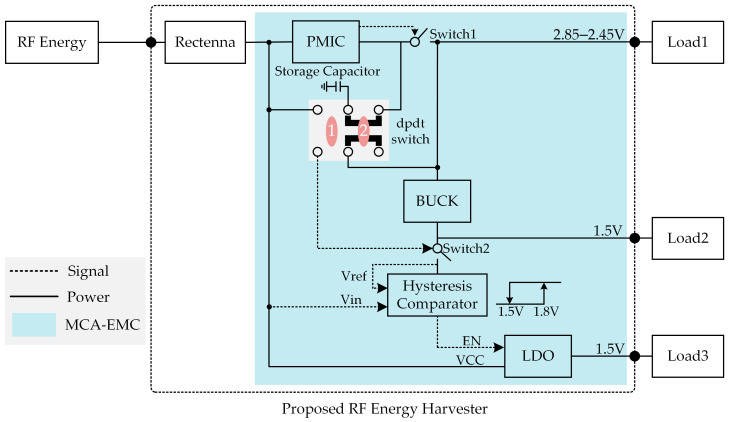
Structure of the proposed RF energy harvester with multi-condition adaptive energy management circuits (MCA-EMC).

**Figure 9 micromachines-14-01967-f009:**
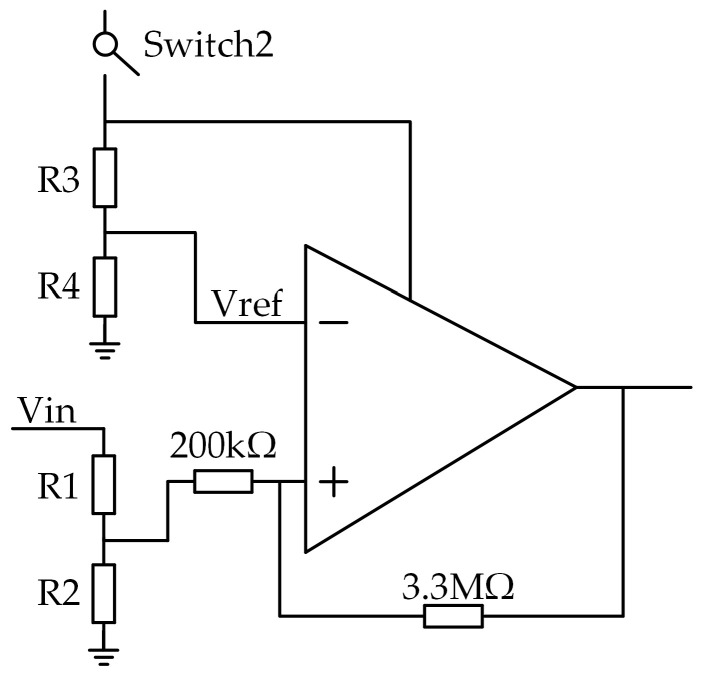
Hysteresis comparator circuits.

**Figure 10 micromachines-14-01967-f010:**
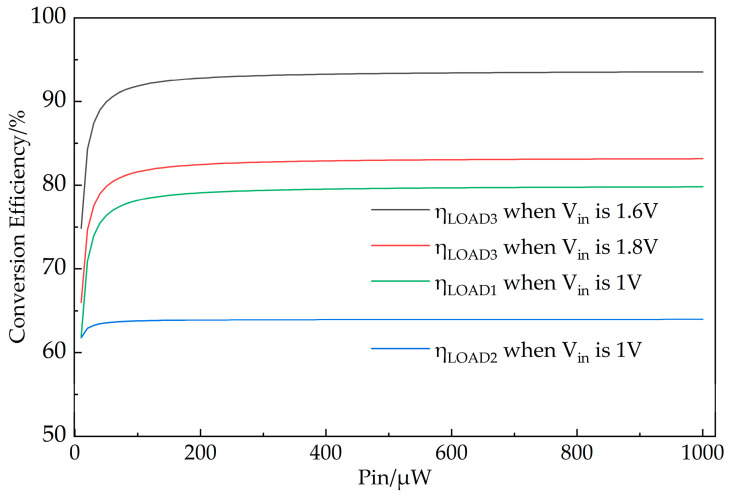
Conversion efficiency of MCA-EMC for three types of loads based on different input voltages versus input power.

**Figure 11 micromachines-14-01967-f011:**
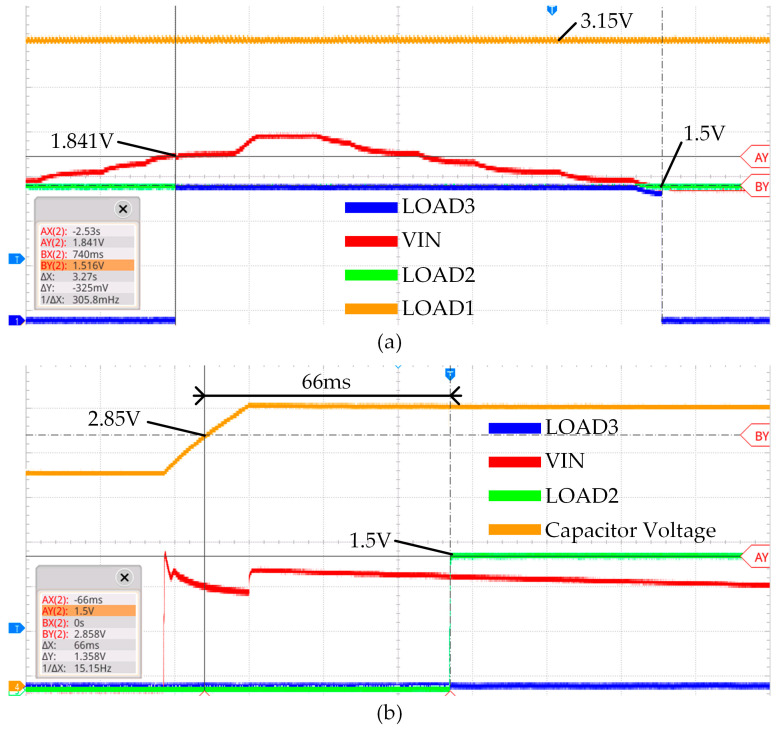
MCA-EMC operational test under different modes (**a**) Mode 1 (**b**) Mode 2.

**Figure 12 micromachines-14-01967-f012:**
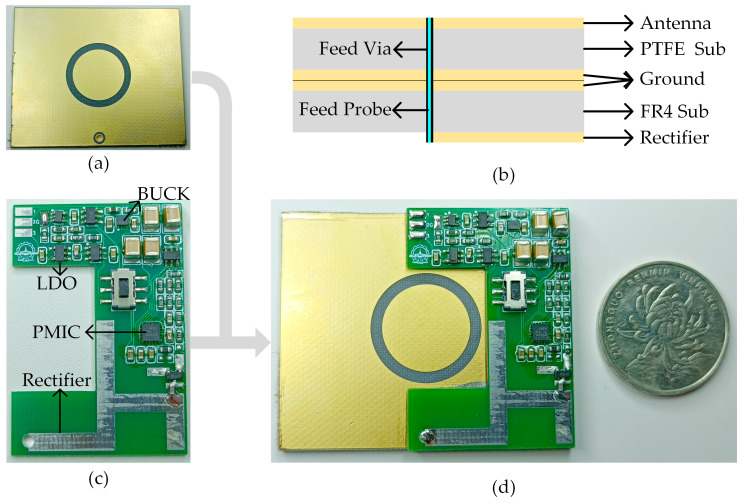
(**a**) Microstrip antenna; (**b**) stacked structure; (**c**) rectifier and the MCA-EMC; (**d**) proposed RF energy harvester.

**Figure 13 micromachines-14-01967-f013:**
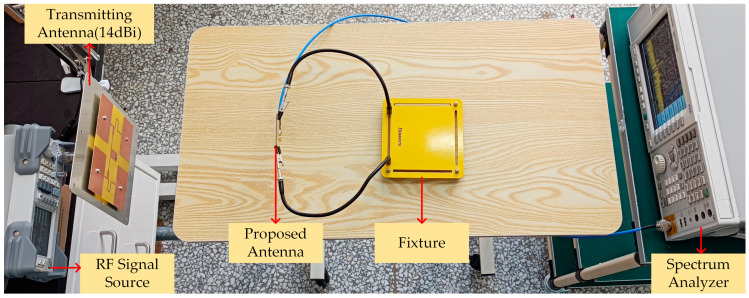
RF energy harvesting and radiation power density test.

**Figure 14 micromachines-14-01967-f014:**
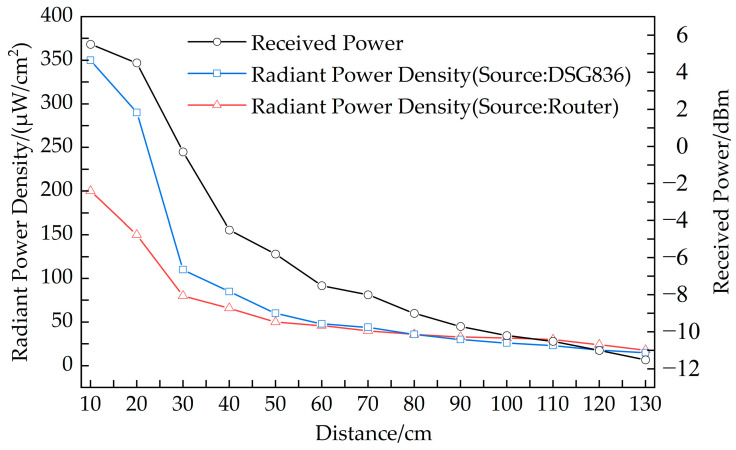
The received power and the radiation power density at the receiving location versus distance between the transmitting antenna and the proposed antenna (as the receiving antenna).

**Figure 15 micromachines-14-01967-f015:**
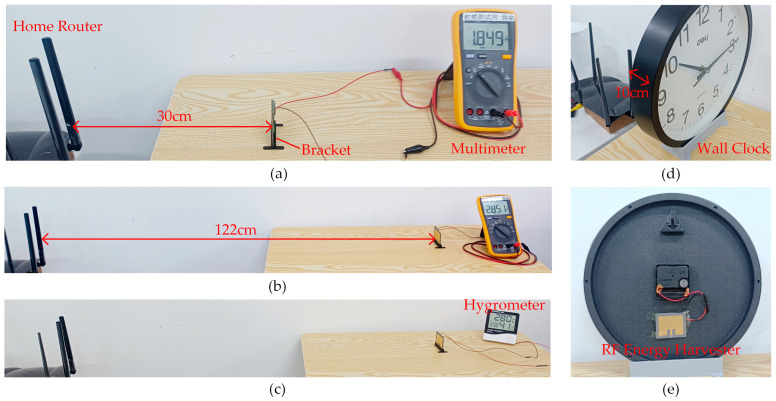
Ambient RF energy harvesting test. (**a**) Maximum working distance test in Mode 1; (**b**) maximum working distance test in Mode 2; (**c**) ambient RF energy drives a hygrometer; (**d**) ambient RF energy drives a wall clock; (**e**) connection of the wall clock to the RF energy harvester.

**Table 1 micromachines-14-01967-t001:** Dimensional parameters of the antenna and rectifier.

**Parameters of the Antenna**	**W0**	**L0**	**W1**	**L1**	**W2**	**L2**	**Wsub**	**Lsub**	**R1**	**R2**
Value (mm)	43	33.5	4.15	15	5	10	53	43.5	8	10
**Parameters of the Rectifier**	**W3**	**L3**	**L4**	**L5**	**L6**	**L7**				
Value (mm)	3.05	11.5	6.45	12.57	8.6	6.41				

**Table 2 micromachines-14-01967-t002:** Leakage current, leakage power, and conversion efficiency (for power chips only) of each module.

	Leakage Current/nA	Leakage Power/nW	Conversion Efficiency (*η*)
LDO	3	3 × *V_in_* (*P_qLDO_*)	VLOAD3/*V_in_* (*η_LDO_*)
BUCK	60	60 × *V_LOAD_*_1_ (*P_qBUCK_*)	0.8 (*η_BUCK_*)
PMIC	330	330 × *V_in_*	0.8 (*η_PMIC_*)
R1 + R2	*V_in_*/(*R*1 + *R*2)	*V_in_* × *V_in_*/(*R*1 + *R*2) (*P_R_*_1+*R*2_)	
R3 + R4	*V_in_*/(*R*3 + *R*4)	*V_in_* × *V_in_*/(*R*3 + *R*4) (*P_R_*_3+*R*4_)	
Hcomp	150	150 × *V_LOAD_*_2_ (*P_HComp_*)	

**Table 3 micromachines-14-01967-t003:** Comparison of the proposed RF energy harvester and related designs.

Ref. (Year)	Total Size	Maximum Conversion Efficiency of Rectifier (%)	Maximum Conversion Efficiency of Energy Management Circuits (%)	Output Voltage (V)
[7] (2021)	−	25% at 1 dBm	82%	3−4 V
[17] (2021)	125 mm × 140 mm	About 10% at −2 dBm	−	1.5−2.7 V
[19] (2019)	150 mm × 90 mm × 50 mm	About 25% at (1 μW/cm^2^)	−	2.3−5.25 V
[14] (2011)	60 mm × 75 mm (antenna) − (rectifier)	55% at (200 μW/cm^2^)	70%	2.5 V
This work (2023)	53 mm × 43.5 mm × 5.9 mm	33.8% at 5 dBm	93.56%	1.5 V, 2.85−2.45 V

## Data Availability

Not applicable.

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
