# Peer review of "A Compact Stacked RF Energy Harvester with Multi-Condition Adaptive Energy Management Circuits"

_micromachines, 2023, doi:10.3390/mi14101967_

Round 1

Reviewer 1 Report

This paper presents a compact stacked RF energy harvester operating in the WIFI band with multi-condition adaptive energy management circuits (MCA-EMC). The test results show that the proposed RF energy harvester can drive a wall clock at 10cm distance and a hygrometer at 122cm distance with a home router as the transmitting source. Overall, this study is interesting. There are some issues that the author should address before publishing:

1. In the introduction, when describing the background of domestic and foreign research, it is not enough to simply cite examples. It is necessary to discuss and evaluate the research appropriately.

2. In the introduction, it is recommended to discuss the areas that need improvement in current RF energy harvesters and propose methods and measures to improve these aspects. For example, mechanical intelligence methods play an important role in improving the performance of energy collection systems. Nano Energy, 2023,108630; Advanced Energy Materials, 2023, 2300557; Nano Energy, 2023, 108222.

3. The device structure mentioned in the article is compact, and the reviewer suggests making a comparison with other studies on device size, energy conversion efficiency, and output power, etc.

4. The formatting of the entire article needs to be further improved.

5. To illustrate the importance of radio-resonant energy harvesting, it is recommended to include comparisons with other mechanical energy harvesters such as wave energy, wind energy, and roadway energy in the article. Nano Energy, 2023, 108222; Sensors and Actuators A: Physical, 2023, 114190; Applied Energy, 2022, 314, 118983.

NO

Author Response

Thank you very much for taking the time to review this manuscript.Please see the attachment. The point-by-point responses and the revised paper have been compiled into a PDF file and uploaded. Please see the attachment.

Reviewer 2 Report

Please see my comments (it has been attached)/ Thanks.

Minor corrections.

Author Response

(The authors gave the same response as above.)

Round 2

Reviewer 2 Report

Thanks.